# Triple Negative Breast Cancer: A Review of Present and Future Diagnostic Modalities

**DOI:** 10.3390/medicina57010062

**Published:** 2021-01-12

**Authors:** Sylvia Annabel Dass, Kim Liu Tan, Rehasri Selva Rajan, Noor Fatmawati Mokhtar, Elis Rosliza Mohd Adzmi, Wan Faiziah Wan Abdul Rahman, Tengku Ahmad Damitri Al-Astani Tengku Din, Venugopal Balakrishnan

**Affiliations:** 1Institute for Research in Molecular Medicine, Universiti Sains Malaysia, USM, Penang 11800, Malaysia; das.viaman@gmail.com (S.A.D.); kimliu@student.usm.my (K.L.T.); srehasri@gmail.com (R.S.R.); 2Institute for Research in Molecular Medicine, Universiti Sains Malaysia, Kubang Kerian, Kelantan 16150, Malaysia; fatmawati@usm.my (N.F.M.); elis@usm.my (E.R.M.A.); 3Department of Pathology, School of Medical Sciences, Health Campus, Kubang Kerian, Kelantan 16150, Malaysia; wfaiziah@usm.my; 4Breast Cancer Awareness & Research Unit, Hospital Universiti Sains Malaysia, Kubang Kerian, Kelantan 16150, Malaysia; damitri@usm.my; 5Chemical Pathology Department, School of Medical Sciences, Health Campus, Kubang Kerian, Kelantan 16150, Malaysia

**Keywords:** triple negative breast cancer, future diagnosis, breast cancer

## Abstract

Triple-negative breast cancer (TNBC) is an aggressive breast type of cancer with no expression of estrogen receptor (ER), progesterone receptor (PR), and human epidermal growth factor receptor-2 (HER2). It is a highly metastasized, heterogeneous disease that accounts for 10–15% of total breast cancer cases with a poor prognosis and high relapse rate within five years after treatment compared to non-TNBC cases. The diagnostic and subtyping of TNBC tumors are essential to determine the treatment alternatives and establish personalized, targeted medications for every TNBC individual. Currently, TNBC is diagnosed via a two-step procedure of imaging and immunohistochemistry (IHC), which are operator-dependent and potentially time-consuming. Therefore, there is a crucial need for the development of rapid and advanced technologies to enhance the diagnostic efficiency of TNBC. This review discusses the overview of breast cancer with emphasis on TNBC subtypes and the current diagnostic approaches of TNBC along with its challenges. Most importantly, we have presented several promising strategies that can be utilized as future TNBC diagnostic modalities and simultaneously enhance the efficacy of TNBC diagnostic.

## 1. Introduction

### 1.1. Breast Cancer

Breast cancer is a group of cancer cells (malignant tumors) that starts in the breast cells and grows out of control. All breast cancer tumor diagnosis starts with the detection of estrogen (ER), progesterone (PR), and human epidermal growth factor receptor-2 (HER2) receptors using immunohistochemistry (IHC) to differentiate the type of breast cancer [1,2,3]. In general, breast cancers are classified into six different intrinsic subtypes including luminal A, luminal B, HER2 enriched, normal-like, basal-like, and claudin-low based on the presence or absence of the three primary markers (ER, PR, and HER2) [4,5], basal marker (CK5/6, EGFR) [6], and Ki-67 proliferation index [7,8,9]. Ki-67 protein is associated with cell proliferation, in which the increased expression of Ki-67 leads to a higher rate of cell division. Figure 1 indicates breast cancer type classification based on immunohistochemical profile differences.

The term “luminal” is used because this type of breast cancer is present at the luminal (inner) epithelial cells of the breast [4,10]. In addition, luminal A and luminal B tumors express a similar protein known as luminal cytokeratin 8 and 18 [4]. Perou and colleagues first discovered the subtype of luminal breast cancer, covering a significant portion of ER-positive immunohistochemical profile in 2000 [4]. In the following year, the luminal subtype was further classified as luminal A and luminal B, dependent on the presence of HER2 expression [11]. As illustrated in Figure 1, luminal A subtype is HER2 negative, and luminal B is HER 2 Positive. In addition, the luminal B subtype is known to propagate faster and has a slightly a worse prognosis than luminal A [11].

HER2 enriched breast cancer subtype is ER and PR negative but HER2 positive. This subtype is known to have faster growth and worse prognosis than the luminal subtype [12]. However, HER2 enriched breast cancer is often successfully treated with Trastuzumab [13]. Typical breast cancer is classified based on the gene expression similarities with epithelial and non-epithelial cells and adipose tissues [14]. The normal-like tumor cells have a low percentage of tumor cells and a lack of proliferation gene expression [10].

The basal-like subtype is ER, PR, and HER2 negative, known as triple-negative breast cancer (TNBC). The term basal-like is contributed by the similarity in the expression of epidermal growth factor receptor (EGFR), CK5/6, CK14, and CK17 [3,4,15,16,17]. TNBC is common among women having a mutation in the breast cancer gene (BRCA) 1 gene [18]. However, not all TNBC are known to be basal-like breast cancer and vice versa [19,20]. Furthermore, Prat and team have indicated that only 70–80% of TNBC are categorized as basal-like subtype [21]. Other factors that can cause TNBC are genetic, signaling pathways, obesity, menopause, multi-party, socioeconomic, and non-breastfeeding [22,23]. TNBC does not respond towards targeted or hormone therapy because of a lack of ER, PR, and HER2 expression. Therefore, cytotoxic drugs are the sole option for TNBC treatment [19].

Claudin-low breast cancer subtype is another intrinsic type identified by their gene expression profiling and is also known as triple-negative breast cancer [20,24]. These tumors are characterized by low E-cadherin expression, mucin-1, epithelial cell adhesion molecule (EpCAM), and claudins (3, 4, 7) [21]. In addition, claudin-low tumors have limited expression of proliferation-associated genes (Ki-67) and luminal markers compared to luminal, HER2-enriched, and basal-like tumors [21]. Moreover, these tumors are known for the elevated expression of epithelial-mesenchymal transition genes (CD14, CD79b, and vav1) and genes involved in cancer stem cell features [22,23,25].

### 1.2. Triple Negative Breast Cancer (TNBC) Subtypes

TNBC is known as a heterogeneous type of cancer that is categorized into six subtypes. The subtypes are immunomodulatory (IM), luminal androgen receptor (LAR), basal-like 1 (BL-1), basal-like 2 (BL-2), mesenchymal (M), and mesenchymal stem-like (MSL) as shown in Figure 2. These types are categorized based upon their gene expression portfolio [26]. For instance, the similarities between BL-1 and BL-2 are the substantial gene expression during cell division as well as cell cycle advancement. Nevertheless, BL-1 retains high gene expression associated with the DNA response pathway, including DNA repair activity and DNA replication, while BL-2 features high expression in growth factor signaling [26]. Alternatively, immunomodulatory (IM) possesses a high expression of genes associated with the immune cell process, specifically a natural killer cell pathway, TH1/TH2 pathway, cytokine signaling, B cell receptor (BCR), and antigen processing. Furthermore, mesenchymal stem-like and mesenchymal subtypes are undoubtedly comprehended to have a high expression of genes associated with extracellular receptor interaction, cell motility, and cell differentiation pathways. Irrespective of that, MSL shows a magnitude difference from mesenchymal subtypes, where MSL has a low expression of claudins (3, 4, 7) genes [26]. Consequently, the MSL subtype is categorized as claudin-low tumors discovered by Herschkowitz et al. [20]. Last of all, the LAR subtype shows high expression in genes related to hormonally regulated pathways and genes regarding androgen receptor as well as its co-activators [26,27]. Nevertheless, Lehmann et al. redefined TNBC molecular subtypes into four tumor-specific subtypes consistng of BL1, BL2, M, and LAR when they found that the IM together with MSL TNBC subtypes were presented from infiltrating lymphocytes and tumor-associated mesenchymal cells [28].

Subtyping TNBC tumors is vital in identifying the treatment alternatives and establishing personalized, targeted medications for every TNBC individual. Table 1 outlines the features of the TNBC subtypes and their respective possible treatment options, as summarized by Lehmann et al. [26,28].

## 2. Current Diagnosis

A two-step procedure typically employed to diagnose TNBC is imaging and immunohistochemistry (IHC) [38]. Imaging encompasses a mammogram, an ultrasound of the breast along with magnetic resonance imaging (MRI) [39]. A mammogram requires a minimal dosage of radiation that does not easily penetrate the breast tissues [40]. Breast cancer diagnosis via mammograms is determined by the presence of calcifications (white spots), growth, or tumor also known as masses [41]. The main challenge is the risk of a false-negative and positive result affecting the diagnosed patient’s treatment outcome [42]. In addition, the side effects of radiation from mammograms may contribute to breast cancer development in high-risk individuals like BRCA gene carriers or family history [43]. Finally, mammography effectiveness is operator-dependent, which may interfere with the result of imaging [44].

Diagnosis via ultrasound is performed when a lump or swelling is not detected in a mammogram but still can be felt and serve as the primary approach to distinguish between breast cysts (fluid-filled sac) and tumors if sample collection is carried out in the right area and tested for cancer [45]. What differentiates breast cysts from a solid tumor is that breast cysts are most often benign, whereas a solid tumor requires further validation to characterize its malignancy [46].

Breast cancer diagnosis by MRI, on the other hand, is opted when a patient is categorized as high risk (family history/BRCA gene mutation) and to determine the severity of the carcinoma due to the efficiency of MRI to detect the early formation of breast cancer in comparison to breast ultrasound and mammogram [47,48]. The main downside of MRI is that the imaging method cannot characterize the breast cancer types and can only confirm the presence of cancer in the breast [49].

Ideally, IHC is required for breast carcinoma typing performed by cell staining with biomarkers such as hormone receptor (progesterone receptor (PR) and estrogen receptor (ER)) as well as human epidermal growth factor receptor two (HER2) markers [50]. In order to enhance the efficacy and accuracy of IHC testing for ER HER2 and PR, there are approximately 126 latest guidelines that have been provided by the American Society of Clinical Oncology (ASCO)/College of American Pathologists (CAP) [51]. The primary aim of these guidelines is to improve the reliability, reproducibility and to reduce the frequency of false-positive and false negative results from IHC testing [52,53]. Based on the recommendation, IHS testing for ER and PR is classified positive only if immunoreactive cancer cells’ presence accounts for a minimum value of 1% [52]. Next, a second confirmation of HER2 should be conducted via fluorescent in situ (FISH) after initial IHC confirmation to obviate any potential false-positive/false-negative diagnosis that will affect the treatment’s direction and effectiveness plan [54].

## 3. Future Diagnosis

### 3.1. Blood-Based Liquid Biopsy

Blood-based liquid biopsy is a non-invasive diagnostic method that can be utilized for future TNBC diagnosis. Liquid biopsy captures the information of a tumor through blood specimen, which is analyzed for the presence of circulating tumor cells (CTCs), tumor-derived extracellular vesicles (exosomes), and circulating tumor nucleic acids (ctNAs), which include circulating tumor DNA (ctDNA) and microRNAs (miRNAs) [55,56]. Based on a similar approach, serum apolipoprotein C-I (apoC-I) has been demonstrated as a potential diagnostic and prognostic marker for TNBC by Song et al. [57].

#### 3.1.1. Circulating Tumor Nucleic Acids (ctNAs)

Analysis of ctNAs include circulating tumor DNA (ctDNA), microRNA (miRNA), and cell-free RNA (cfRNA) [58]. CtDNAs found in the bloodstream of a cancer patient is usually from the primary tumor [59,60], CTCs [61], and from apoptotic and necrotic cell deaths during cancer development and progression [62,63,64]. The volume of tumor ctDNAs in the bloodstream depends on the size of the tumor or metastases, and a study has concluded that the ctDNA concentration will increase the percentage of tumor burden [65]. Hence, it is difficult to detect ctDNA in an early stage of cancer as only a low concentration of ctDNA can be found. This suggests that an ultrasensitive technology is urgently needed to detect the initial stage of cancer as the levels of ctDNA present are low. One such technology is the droplet digital polymerase chain reaction (ddPCR), which was able to detect phosphatidylinositol-4,5-biphosphate 3-kinase catalytic subunit alpha (PIK3CA) mutations in the blood specimen of early-stage breast carcinoma patients [66]. However, further validation and development are necessary before ctDNA can be utilized as a biomarker for early breast carcinoma diagnosis. On the other hand, assessing ctDNAs in the plasma can be used for real-time monitoring of the tumor burden and measure the effectiveness of treatment [65]. This is due to the fact that ctDNAs have a short half-life (15 min to several hours) [67,68], allowing earlier observation of ctDNA level changes in the bloodstream than radiological images. Moreover, ctDNA analysis can be an alternative to confirm the diagnosis of metastatic relapse, which was evident in a study that has demonstrated that ctDNA analysis was capable of detecting early disease metastasis of a patient who underwent treatment for early breast carcinoma [69]. However, ctDNA’s prognostic value is still under assessment, although Chen and colleagues have shed some light on the prognosis of ctDNA in determining the status of the cancer post neoadjuvant chemotherapy [70].

MicroRNAs (miRNAs) are short ribonucleic acids (RNAs) made up of approximately 22 nucleotides that regulate thousands of genes via binding to target messenger RNAs (mRNAs) [71]. miRNAs play various roles in many biological processes such as cell development, growth, differentiation, chromatic structure, cell death, metabolism, and morphogenesis [72,73,74]. In addition, miRNAs that also act as oncogenic miRNAs or tumor suppressors play an essential role in tumorigenesis [74,75]. Oncogenic miRNAs were used to demonstrate anti-apoptotic activity and were found to be overexpressed in cancer cells [76,77,78]. In contrast, tumor suppressor miRNAs usually display anti-proliferative, pro-apoptotic activity, and downregulated in cancer cells [79,80].

A study by Thakur et al. indicated a high expression of miR-21, miR-220, and miR-221 in TNBC Indian women [81], which reconfirms the findings of Radojici et al. [82]. In contrast, the expression of miR-21 and miR-221 was downregulated in a study based in Hong Kong, highlighting the possibility of miRNA expression variation in distinct ethnic groups or the geographic location of the patients [83]. Moreover, several other non-TNBC specific studies showed different expression levels of miR-(21,221,195,145) and Let-7a in other types of breast carcinoma categories [84,85,86,87]. This suggests that the expression level of miRNA depends not just on the tumor type but also on the breast cancer stage and grading. Thus, Frères et al. had developed a new screening tool for breast cancer by constructing a diagnostic test based on eight circulating miRNAs (miR-(16, 103, 107, 148a, 19b, 22) and let-7(d and i) [88]. The authors were able to prove that the newly developed method was able to identify breast carcinoma malignancy as well as early detection of breast cancer incidences. The method produces results independent of the age and tumor stage of the patient. However, the sampling population is only limited to local patients and cannot classify different types of breast cancer. Therefore, the method needs to be validated in other regions and further developed to identify the types of breast cancer.

#### 3.1.2. Exosomes

Reported initially by Pan and Johnstone in 1983, exosomes are extracellular, membrane-bound vesicles that are secreted by many cells under normal and abnormal circumstances [89]. The exosomes primarily involve transporting biomolecules, including DNA, RNA, proteins along with lipids to recipient cells [90,91]. In addition, exosomes also play a role in cell signaling and intercellular molecular communication [92,93]. A study by O’Brien et al. has demonstrated the ability of exosomes from TNBC to aid in communication between cells as well as phenotypic traits transfer to the secondary cells [94].

During carcinogenesis, exosomes from the cancer cells were found to trigger cancer cell proliferation and stage immune defense escape ultimately promoting cancer progression and metastasis [95,96]. In a study by Piao et al., the exosomes from TNBC were found to cause tumor growth and lymph node metastasis via intercellular communication with macrophages [97]. Several studies have shown that exosomal proteins can be used as diagnostic and prognostic markers. Rupp et al. had shown that CD24 might serve as a circulating breast cancer biomarker, although CD24 can also be found in numerous cancer types such as colorectal cancer [98]. In addition, Moon et al. has suggested that endothelial Locus-1 (Del-1) and fibronectin of the circulating exosomes from the plasma can be considered as potential biomarker candidates for early detection for breast cancer patients [99]. Nonetheless, the finding is not specific to TNBC, but the breakthrough of the findings may serve as an essential guideline for future development in the diagnosis of TNBC. In conclusion, liquid biopsy provides real-time, reliable results, reduces the cost and diagnosis time, and allows patients to avoid the risk of surgery.

### 3.2. Immuno Positron Emission Tomography (PET)

Positron emission tomography, also known as PET scan, is a medical imaging approach that utilizes a radioactive element/drug to analyze the organ and tissue functionality and is well-known for its capability to detect a particular disease even before detection by other imaging methods [100]. In this approach, the radioactive element (tracer) is comprised of tightly linked radioactive atom-transport molecules (isotopes) that adhere to specific biomolecules (sugar, protein, etc.) in the human body and generate positrons that inter-reacts with the surrounding electrons resulting in the formation of photons [101]. The PET scanner then detects the electrical signal emitted by the photons and utilizes the data obtained to generate the image of the organ/tissue/cell being investigated [102].

Based on a similar approach, immune-PET imaging utilizes the integration of the PET system along with monoclonal antibodies (mAbs) to improve the efficacy of tumor characterization diagnosis and aid in selecting suitable targeted mAb-based therapy [103]. In this approach, the antibody’s primary role is to locate specific cell surface tumor markers or components located at the extracellular matrix, which is then recognized by the PET detection system [104]. Prominent evidence of this concept is the discovery of ATL-836 fragment antigen-binding (Fab) chimeric monoclonal antibody against human tissue factor (TF) [105]. The discovery of ATL-836 antibody provides a promising platform for future diagnostics and therapeutics of TNBC as TF, otherwise known as platelet tissue factor/factor III, engages a crucial role in the signaling of cancer cells (apoptosis inhibition and cell migration promotion) and is found to be prominently presented on TNBC cells [106,107]. In 2017, another promising TNBC diagnostic imaging antibody agent targeting glycoprotein non-metastatic B (gpNMB)/osteoactivin was successfully developed in a TNBC xenograft animal model [108]. This discovery is crucial as gpNMB expression is significantly high in TNBC patients and, most importantly, in tumor progression reoccurrence [109,110]. In addition, the conjugation of the antibody to the toxin was capable of inhibiting the proliferation of gpNMB-expressed TNBC cells [110,111]. In a nutshell, immuno-PET imaging is not only about early detection of TNBC, but it is also able to identify the suitable therapeutic method for a patient because immuno-PET is able to image the expression of therapeutic targets.

### 3.3. Nanobiosensor

A biosensor is a tool comprised of bioreceptor, detector, and the signal transducer, utilized for the identification and analysis of a wide range of biological specimen, including enzymes, immune components (antigen and antibodies), nucleic acid components (DNA, RNA, microRNAs, and ctDNA), and other biological components present in humans [112]. Bioreceptor is an immobilized sensitive biological element (enzyme, DNA probe, antibody) recognizing the analyte (enzyme substrate, complementary DNA, antigen). Transducer is used in biosensor to convert (bio) chemical signal emitted from the interaction of analyte with bioreceptor into an electronic signal. The intensity of the generated signal is directly or inversely proportional to the analyte concentration. Electrochemical transducers are often used in biosensors [113]. Biosensors are categorized according to the basic principles of signal transduction and biorecognition elements. According to the transducing elements, biosensors can be classified as electrochemical, optical, piezoelectric, and thermal sensors. Electrochemical biosensors are also classified as potentiometric, amperometric and conductometric sensors [114]. Although antibodies and oligonucleotides are widely employed, enzymes are by far the most commonly used biosensing elements in biosensors.

Recognition begins when the bioreceptor binds to a distinctive biological analyte, which generates measurable binding signals by signal transducer and finally detected by the detector for data analysis [115]. Nanobiosensor, as the name suggests, is a biosensor integrated with nanoparticles combined with transducers that enhance the biological signaling and transduction process (Figure 3) [116]. This is possible as the nanoparticles demonstrate (high surface area: volume) ratio contributed by its small size, which subsequently amplifies the sensor’s receptivity and reduces the detection cut-off point by recognizing biological analytes of low concentrations.

In terms of TNBC cell detection, several nanobiosensors have been developed in the past. The zinc oxide (ZnO)-choline oxidase (ChOx) nanobiosensor generated in 2016 was able to identify the presence of choline in TNBC samples [117]. In another approach, an electrochemical-nanobiosensor designed based on the LNA oligonucleotide probe exhibited promising TNBC diagnostic potentials by successfully recognizing tumor-associated miR-199a-5p marker (detection limit of 4.5f), which in general was found to be downregulated in TNBC cells in comparison to normal cells [118,119]. As highlighted above, the nanobiosensor was found to be highly sensitive and selective in detecting the low miR-199a-5p concentration in the patient’s blood specimen. Other findings include a dual-ligand co-functionalized gold nanocluster (AuNCs) with the ability to recognize as well as distinguish between carcinoma, non-carcinoma (normal), and metastatic breast cancer cells, which also take into account the TNBC cells, highlighting its promising analysis and diagnostic potentials of the nanobiosensor [120].

### 3.4. nCounter^®^ Breast Cancer 360™ Panel

The nCounter^®^ Breast Cancer 360™ (Seattle, WA, USA) Panel initiated in April 2018 is an analytical data tool comprising approximately 770 genes to aid in breast carcinoma classification based on molecular subtyping [121]. In this diagnostic method, the patient’s RNA sample is extracted and integrated overnight with the Breast Cancer 360^TM^ panel assay before performing specimen and data analysis using the Nanostring nCouter^®^ system (Seattle, WA, USA) [122]. The system provides an in-depth understanding of the level of gene expression, immune defense mechanism towards the breast carcinoma, and tumor microenvironment along with breast cancer categorization formulated on biological signatures such as prediction analysis of microarray 50 (PAM50) and tumor inflammation signature assays [4,123]. This efficiency of NanoString BC360^®^ (Seattle, USA) was evident in Phase I clinical trial evaluating Eribulin and Everolimus in TNBC candidates whereby the panel was capable of disclosing the diversity of breast cancer and its microenvironment [124]. In another study, the NanoString^®^BC360 panel aided in distinguishing the intrinsic breast carcinoma subtypes and subsequently evaluated endocrine therapy effectiveness for stage I luminal breast cancer [125]. In addition, the validity of NanoString BC360^®^ in determining breast cancer subtype (ESR1, PGR, MK167, and ERBB2 genes) was recently proven to draw a parallel with the traditional immunohistochemistry method [126]. In general, the panel lacked in terms of calling for a substantial number of samples for data validation, and it has been only applicable for only research usage. It is to be expected that NanoString BC360^®^ panel will be utilized for breast cancer diagnostics in the future.

### 3.5. Digital Polymerase Chain Reaction (dPCR)

Introduced by Vogelstein and Kinzler in 1999, digital PCR is a method that segregates the samples into multiple wells before the amplification process [127]. Figure 4 below illustrates the overview of dPCR.

The pros of dPCR compared to a conventional quantitative polymerase chain reaction (qPCR) are that there is no requirement for a standard curve for analysis, it is able to tolerate any PCR inhibitors [128], able to analyze the presence of uncommon targets in large sample mixture, and capable of identifying minute fold changes [129]. In addition, sample segregation and absolute quantification in dPCR allow it to be suitable detection candidates of rare allele [130,131], genomic mutations such as variation, DNA deletions or replication [132,133,134], viral load, and next-generation sequencing libraries quantification [129,135,136,137,138,139,140]. In general, digital PCR is utilized for circulating tumor DNA and miRNA identification in cancer patients [66,141,142]. In 2019, a 4-plex droplet digital PCR (ddPCR) was designed for the simultaneous analysis of four breast cancer oncogenes (PUM1, ESR1, PGR, and ERBB2) to determine the breast cancer subtype [143]. Currently, there are several commercial dPCR (Raindrop™ Digital PCR System (Raindance™ Technologies, Billerica, MA, USA), QX100™ and QX200™ Droplet Digital™ PCR System (Bio-Rad, Hercules, CA, USA), BioMark™ HD System and qdPCR 37K™ IFC (Fluidigm Corporation, South San Francisco, CA, USA), QuantStudio™ 3D Digital PCR System (Life Technologies™, Carlsbad, CA, USA) and Clarity (JN Medsys, Singapore, Singapore) available for diagnosis [144]. In short, digital PCR provides a promising platform with great accuracy for the early detection of cancer.

In general, all three diagnostic methods discussed above are based on the presence and expression of specific genes by the cancer cells. Hence, a summary of TNBC classification based on gene expression profiling (Table 2) would provide a useful platform for developing and applying a suitable diagnostic approach for the diagnostics of TNBC.

## 4. Conclusions

Triple-negative breast cancer (TNBC) is an aggressive type of cancer but lacks targeted therapy methods such as hormone therapy due to the low expression of three primary receptors (ER, PR, and HER2). Therefore, novel methods that can detect TNBC in real-time, accurate, and minimally invasive ways are urgently needed. This ensures that proper treatment can be provided in the early stages of cancer, and the treatment’s efficiency can be monitored.

## Figures and Tables

**Figure 1 medicina-57-00062-f001:**
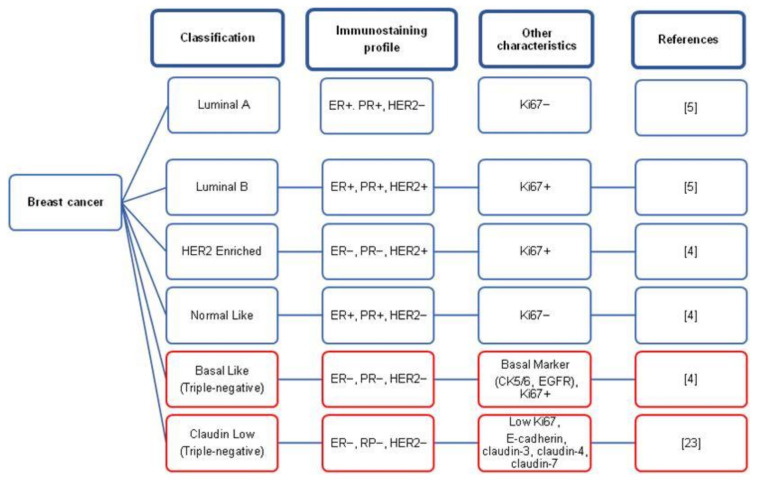
The intrinsic subtyping of breast cancer based on immunohistochemical profile.

**Figure 2 medicina-57-00062-f002:**
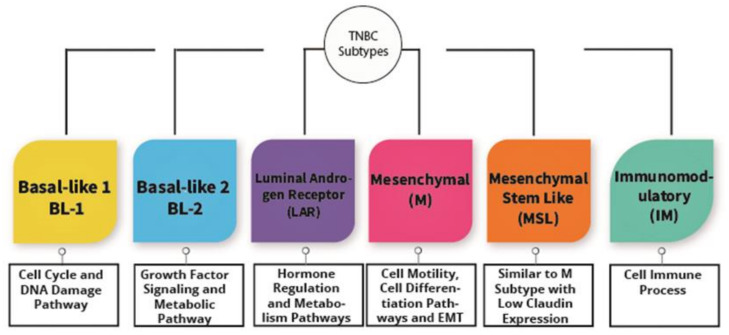
The subtypes of triple negative breast cancer.

**Figure 3 medicina-57-00062-f003:**
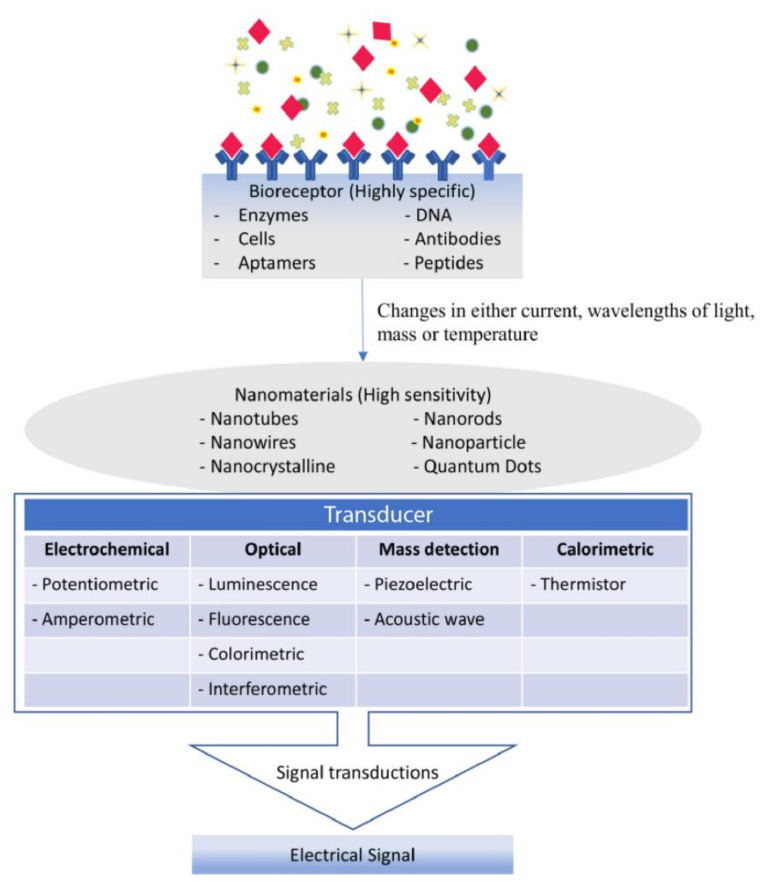
Working principles of nanobiosensor. Bioreceptor form biological reaction when bind to sample analyte which caused changes in either current, wavelength, mass, or temperature. The biological response will then be converted to electrical signals via the transducer. Nanomaterials that integrate with transducers is to detect low concentrations of analytes.

**Figure 4 medicina-57-00062-f004:**
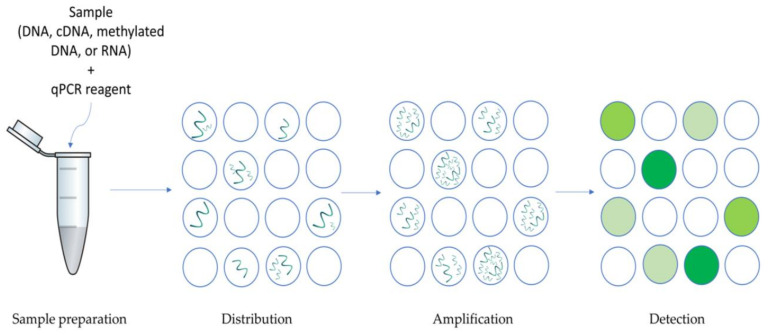
Overview of dPCR. Sample is added and combined with qPCR reagent. It is then equally distributed into many sub-volumes (either in microwells, chambers, or droplets) which results in some partitions to contain few targets and some without. Each subdivided portion will undergo amplification. The positive columns will be identified and the target concentrations will be determined via Poisson’s statistics.

**Table 1 medicina-57-00062-t001:** Characteristics and possible treatment options based on TNBC molecular subtypes. Abbreviations: PARP: poly-ADP ribose polymerase; AR: androgen receptor; Wnt: Wingless-related integration site; PI3K: phosphoinositide 3-kinase; mTOR, mechanistic target of rapamycin; TGF-β, transforming growth factor beta.

TNBC Type	Characteristics	Treatment Options
Basal-like 1 (BL-1)	DNA damage response pathway	PARP inhibitors [29] Platinum compounds [30,31]
Basal-like 2 (BL-2)	Growth factor signaling, glycolysis and gluconeogenesis	Growth signaling inhibition [32]
Luminal Androgen Receptor (LAR)	High expression of genes related to hormone	AR antagonists [33]
Mesenchymal (M)	Cell differentiation pathway, interaction between extracellular receptor, mobility of cell	Wnt/β-catenin inhibitors [34] PI3K/mTOR inhibitors [26,35] TGF-β receptor kinase inhibitors [36]
Mesencymal Stem Like (MSL)	Similar to M subtype but is claudin-low and high expression of mesenchymal stem cells	
Immunomodulatory (IM)	Immune cell process	Immune check point Inhibitors [37]

**Table 2 medicina-57-00062-t002:** List of elevated gene expression based on TNBC subtypes.

TNBC Type	Characteristics	Elevated Gene Expression
Basal-like 1 (BL-1)	Cell cycle and cell division components and pathways	AURKA [145], AURKB [146], CENPA [147], CENPF [148,149], BUB1 [150,151], TTK [152], CCNA2 [151], PRC1 [153] MYC [154,155], NRAS [156,157], PLK1 [157], and BIRC5 [151]
	DNA damage response pathway	CHEK1 [26,158], FANCA [26,159], FANCG [26], RAD54BP [26], RAD51 [26], NBN [26], EXO1 [26,160], MSH2 [26], MCM10 [26], RAD21 [26], and MDC1 [26]
Basal-like 2 (BL-2)	Growth factor signaling	EGFR [161,162], MET [163], and EPHA2 [164]
Immunomodulatory (IM)	Immune cell processes	TH1/TH2 pathway [165], NK cell pathway [165], B cell receptor (BCR) signaling pathway [165], DC pathway [165], T cell receptor signaling [165], cytokine pathway [165], IL-12 pathway [165], IL-7 pathway [165], NFKB [166], TNF [166] and JAK/STAT signaling [166].
Mesenchymal (M)	Cell motility, extracellular receptor interaction, and cell differentiation pathways	TGFB1L1, BGN SMAD6, SMAD7, NOTCH1, TGFB1, TGFB2, TGFB3, TGFBR1, TGFBR2, TGFBR3, MMP2, ACTA2, SNAI2, SPARC, TAGLN, TCF4, TWIST1, ZEB1, COL3A1, COL5A2, GNG11, ZEB2, FGF, IGF, PDGF, CTNNB1, DKK2, DKK3, SFRP4, TCF4, CF7L2, FZD4, CAV1, CAV2, and CCND2 [26]
Mesencymal Stem Like (MSL)	Similar to M subtype but is claudin-low and a high expression of mesenchymal stem cells	VEGFR2 [167], TIE 2 and TIE1 [168], KDR, EPAS1, ABCA8, PROCR, ENG, ALDHA1, PER1, ABCB1, TERF2IP, BCL2, BMP2, THY1, HOXA5, HOXA10, MEIS1, MEIS2, MEOX1, MEOX2, MSX1, BMP2, ENG, ITGAV, KDR, NGFR, NT5E, PDGFRB, THY1, VCAM1, FBN1, MMP2, PDGFRB, THY1, SPARC, TGFBR2, PDGFRA, TWIST, CAV1, CAV2, and SERPINE1 [169]
Luminal Androgen Receptor (LAR)	High expression of genes related to hormones	DHCR24, ALCAM, FASN, FKBP5, APOD, PIP, SPDEF, CLDN8, FOXA1, KRT18, XBP1 [26]

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
