# Peer review of "Triple Negative Breast Cancer: A Review of Present and Future Diagnostic Modalities"

_medicina, 2021, doi:10.3390/medicina57010062_

Round 1

Reviewer 1 Report

The concept of such a review is very appealing, and I appreciate how it is well structured. Logic is straightforward, and breaking information into sections delivers the data well. The figures are demonstrative and useful.

However, the many errors, typos and inaccuracies in the text (including inaccuracies in scientific terms) makes it impossible to be published in present form. After the text of this manuscript is polished, the idea of publishing this review is very welcome.

Here are my main concerns and comments regarding the manuscript.

  • “TNBC” is sometimes compared to “breast cancer” throughout first pages of the manuscript. However, TNBC is also breast cancer, so in these cases it’s better to compare it to “non-TN breast cancer” instead of “breast cancer”. For example, in lines 25 and 35.
  • Lines 30-31: ultrasound and MRI are not used to diagnose TNBC but just a fact of tumor/cancer. And I don’t see any evidence they are “hazardous”, as it is stated in the text.
  • Lines 23 and 27: “15-20%....” – repeat.
  • Figure 1: it needs to be reflected on the Fig.1 AND in the text that claudin-low BCs are also TN.
  • Not sure why ER, PR, HER2 expression status is called “immunoprofile”. By the most common method their expression is assessed (IHC)? Immunohistochemistry is just a method, it doesn’t assess any immune features in this case.
  • Line 53: either “luminal epithelia” or “epithelial cells”
  • Throughout the text some words are randomly capitalized, e.g. luminal/Luminal. It will look better if everything is brought to the same appearance.
  • Line 139: is EGFR2 mentioned for the first time here? Please mention that it’s the same that HER2.
  • Lines 139-140: “improvised”: do authors mean “improved”? Also, these lines repeat lines 143-144.
  • Chapter 3.1.1: can it be listed together with other TNBC diagnostics methods while, from what is described, it’s actually a method to predict survival and monitor disease progression (as this review emphasizes TNBC diagnostics itself)? If authors want to describe this one then many others should be mentioned, because there are many more studies working on development of survival prediction, etc. in TNBC.
  • Line 201: microRNAs are not long.
  • Chapter 3.3: The principle of biosensors method is not clear, for example in line 277: what is bioreceptor? an antibody?
  • Chapter 3.5: What is the principle of Prosigna tool? How is it different from PAM50 described in chapter 3.4 (nCounter)?

Even though the listed concerns regarding the manuscript text are important, I still consider them minor. After the text is corrected, it can be published.

Reviewer 2 Report

The authors have presented a review of TNBC, and the current and future mechanisms of its diagnosis. There are comments below that may be considered in order to further improve the quality of the review. 

Minor comment: Too many errors in grammar. The paper should undergo a thorough revision by a scientific/English editor. Some of the errors are highlighted in yellow in the uploaded file.

Title: The current title of this paper conveys little or no meaning to the reader, and should be revised. A more appropriate title would be "Triple negative breast cancer: A review of present and future diagnostic modalities."

Abstract: This is poorly written. Emphasis should be on the currently available diagnostic modalities (and their disadvantages), and future options (and their advantages). The aims of the review should be clearly stated... "overview of breast cancer, TNBC, and present and future diagnostic modalities of TNBC."

In line 21, revise the phrase "Triple-negative breast cancer (TNBC) is known as an aggressive breast cancer type.." to "Triple-negative breast cancer (TNBC) is an aggressive breast cancer type..." Line 27, delete the repeated information "15-20% of a real breast cancer diagnosis. Besides,". Lines 29-30, the sentence does not make sense, and is incorrect; revise: "Usually, cancer diagnosis begins after the cancer cells have metastasized or discovered in the late stage due to misdiagnosis."

The introduction and the rest of the review are well written except for a lot of error in grammar. This should be corrected.

Please find attached a pdf file where highlighted parts need to be revised.

Best of luck!

Round 2

Reviewer 2 Report

The authors have responded adequately to my comments and the I believe that the current  draft of the manuscript is much improved.

Author Response

Below are the complete details on the revision of our manuscript.

  1. "There are some improper citations left. For example, there are the sentences
  2. "Table 1 outlines the features of the TNBC subtypes and their respective possible treatment options as summarized by Lehmann et al. (2011 & 2016)"

Response: The citations (26,29) have been added (Line 116)

  1. Table 2, some citations are provided like this: "EGFR Changavi et al., 2015; Hashmi et al., 2019)".

Response: The citations (162,163) have been added (Table 2)

  1. Abbreviations have been defined
  2. breast cancer gene (BRCA) Line 69
  3. epithelial cell adhesion molecule (EpCAM) Line 78
  4. phosphatidylinositol-4,5-biphosphate 3-kinase catalytic subunit alpha (PIK3CA)

Line 174

  1. ribonucleic acids (RNAs) Line 186
  2. prediction analysis of microarray 50 (PAM50) Line 306

  1. All the terms were standardized to lower case

Line 85-86